# Integrated vs. Specialized Farming Systems for Sustainable Food Production: Comparative Analysis of Systems' Technical Efficiency in Nebraska

Maroua Afi 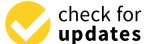 and Jay Parsons *

Department of Agricultural Economics, Institute of Agriculture and Natural Resources,
University of Nebraska-Lincoln, Lincoln, NE 68588, USA
* Correspondence: jparsons4@unl.edu; Tel.: +1-402-472-1911

**Abstract:** Complementarities between crops and livestock production have the potential to increase input use efficiency and maintain a diversified livelihood. This paper uses non-parametric data envelopment analysis (DEA) to assess the technical efficiency (TE) of integrated crop–livestock systems (ICLS) compared to specialized cropping and specialized livestock systems in the state of Nebraska, in the central United States. We classify each county of Nebraska into one of three systems according to their dominant agricultural production revenues. We use DEA to measure the TE of each county compared, first, to a group production frontier (in-system comparison) and second, to a metafrontier (cross-system comparison). Thirty percent of the cropping systems counties were evaluated as fully efficient in the in-system comparison with other cropping systems counties. Thirty-six percent of the livestock systems counties and 18% of the ICLS counties were evaluated as fully efficient in their in-system comparisons. The ICLS counties are less likely to appear on the metafrontier, with a total of only 7% compared to 39% and 32% for the specialized cropping and specialized livestock systems, respectively. These results highlight the need for further research on optimal crop–livestock integration that allows for the realization of synergies and complementarities needed for higher efficiency and sustainable intensification of food production.

**Keywords:** integrated crop–livestock systems; specialized systems; technical efficiency; metafrontier; sustainable food production

## 1. Introduction

Sustainable intensification of land use practices is promoted to address an increasing food demand and to ensure food security for a growing world population. "Sustainable intensification" is defined as "producing more outputs with more efficient use of all inputs–on a durable basis–while reducing environmental damage and building resilience, natural capital, and the flow of environmental services" [1]. Scientists advocate for integration practices over specialization to improve soil health and sustain crop productivity as well as to intensify food production without further compromising natural capital [2]. Complementarities and synergies between crops and livestock enhance nutrient cycling and the delivery of ecosystem services which have long-run positive impacts on yields and productivity [3]. Sustainable intensification of food production is intended to increase resource-use efficiency while reducing agricultural land area expansion [4]. The latter creates an opportunity for the re-integration of crops and livestock enterprises as a promising transition pathway in agricultural production; these systems provide several advantages for nutrient cycling [5], and they deliver ecosystem services by improving ecological processes [6]. The main objective of this paper is to evaluate technical efficiency (TE) of integrated crop–livestock systems (ICLS) compared to specialized systems (SS)—livestock-specialized and crop-specialized. Following the creation of a typology of Nebraska counties divided into three groups according to the omnipresent agricultural production system, this study is based on two analytical

steps. The first step is to perform a non-parametric data envelopment analysis (DEA), allowing for an in-system technical efficiency comparison according to the group frontier. The second step is to create a metafrontier aiming to provide a cross-system comparison. The analysis is based on aggregate county-level data in Nebraska.

Nebraska, located in the Midwestern region of the United States, is known for its diversified landscape and rich natural resources that span from east to west. Being one of the major food suppliers in the U.S. and in the world, Nebraska produces a variety of crops, particularly corn and soybean, in addition to an abundance of beef products. It is also a leading state in plant ethanol production. Integration between crops and livestock in Nebraska has both a spatial and a temporal scope, making it an ideal region to study the potential frameworks of integration allowing for successful diversification. In the upcoming sections of this paper, the counties constitute the decision-making units (DMUs) of the data envelopment analysis applied.

Estimating the technical efficiency using group frontier models is beneficial for creating profiles of efficient and inefficient farming and ranching systems. In our study, we compared each county in Nebraska to other counties in Nebraska with a similar type of agricultural production system. We refer to this as an "in-system comparison". From another point of view, the estimation of the distance between each group frontier and the metafrontier or the "cross-system comparison" allows for a comparison between counties from different groups; for instance, we can compare the efficiency of one county from the specialized cropping system counties to a nearby integrated crop–livestock county. This step helps to determine which production systems across the state are more efficient and to design policies and programs improving the overall production environment [7]. This comprehensive approach provides novel insights into the potential benefits and barriers of integrated crop–livestock systems for achieving sustainability goals in food production, compared to specialized systems. Our findings suggest the need for further research into the optimization of crop–livestock integration that investigates the realization of synergies and complementarities that enhance technical efficiency. Ultimately, this may be a key component in promoting sustainable intensification of food production.

The following sections include an extended literature review on the characteristics of specialized cropping systems, specialized livestock systems and integrated crop–livestock systems. It highlights the synergies, complementarities, and tradeoffs in integrated systems. This is followed by a section presenting the methodological framework and the data, and, finally, we present a results and discussion section including the implication of our study.

## 2. Literature Review

Integration of livestock production into diversified cropping systems is beneficial on the economic and the environmental levels, despite some potential drawbacks such as soil compaction and interference with other crops [8]. Mixed crop–livestock production systems provide a diversified set of outputs and are the main rural livelihood basis for many of the world's rural poor [9]. These systems also provide 50% of global food production [10]. Crop–livestock production combinations provide farmers with various income sources and reduce risk [11]. Livestock are an important stabilizing source of food and income; when crop income faulters, farmers rely on livestock as their alternative source of income. Crops can also provide input to animal production as harvested feed or grazing resources; meanwhile livestock provides manure for crop fertilizer [12]. Thus, livestock sustain soil fertility through excreta and crops sustain livestock through feed sources. ICLS are beneficial for plant growth because the nitrogen produced by legumes such as soybean is better absorbed than the mineral nitrogen supplied through external sources of fertilization, thus enhancing biomass quality [13]. However, specialized livestock systems, based on grass monoculture pastures, have lower dry matter production which diminishes soil moisture, organic carbon, and nitrogen compared to ICLS [14].

Diversified systems have been shown to have several advantages over specialized farming systems in terms of contributing to sustainability goals [6]. Integrated crop–

livestock systems can increase resource use efficiency and limit the harm to the environment by allowing the recycling of nutrient and energy inputs between the two components of the system [9,12,15]. Moreover, diversification of production patterns enhances resource use efficiency and reduces reliance on a single crop or animal enterprise. This diversification can help build resilience and enhance the adaptability of the agricultural production system in the face of environmental and economic pressures [16]. Specialized agricultural production systems are often more vulnerable to environmental variability and economic stress due to dependency on a narrow set of inputs and outputs [16]. Additionally, specialized systems have high synthetic input usage, monoculture practices, and soil disturbance which may engender soil degradation, water pollution, and other negative environmental impacts, and directly compromise sustainability goals in food production.

One of the impediments to this crop–livestock synergy is the inherent nature of mixed systems that entails a competitive usage of crop residues, since these are used as soil cover in addition to serving as animal feed [17]. In addition to the trade-offs in grazing crop stubble, several studies have highlighted grain yield reductions in ICLS [18–21], these reductions are due to the competition between annual crops and pasture during growth season in crop-pasture intercropping. Labor management and qualified labor encompassing the required knowledge to manage such complex systems are also a great challenge for integrated systems. Therefore, to capture all the benefits of ICLS, management decisions should ensure that all trade-offs are balanced and focus on finding and reducing efficiency gaps to achieve the overall objectives of the enterprise.

U.S. agricultural production systems have become increasingly specialized, which has created multiple social and economic benefits. However, this specialization engendered multiple negative externalities on animal welfare [8], and it has also caused environmental degradation and a loss of biodiversity in the United States [22]. Sulc and Franzluebbers (2014) explored how integrated crop–livestock systems would contribute to achieving environmental stewardship while maintaining economic profitability in widely diversified natural and ecological conditions in the United States [23]. The study performed an analysis of the economic performance of integrated systems across the U.S. considering different characteristics of different systems, location, soil types, crop types, livestock species, etc. Meanwhile, the study also investigated the diminishing contribution of agriculture to the GDP (from 8% in 1930 to less than 1% in 2000). It pointed out how agricultural activity is of less importance to policymakers, thus, current agriculture research and extension in the U.S. are not sufficient to explore and expand more sustainable alternatives for food production. The study highlighted that such complex agricultural systems "require greater managerial complexity in the face of more expensive and insecure fossil-fuel supplies, changing and less predictable climate". A statement from which we can deduct the need to conduct more research to increase the understanding of the functioning of such systems and investigate their efficiency against climate and market risks.

In the United States, agriculture activities contribute to 9% of greenhouse gas emissions [24]. The integration of pasture and grazing into specialized cropping systems is anticipated to decrease the emissions and lower the water pollution caused by fertilizer and herbicide application [22]. Hence, it provides a sustainable alternative for intensifying food production and improving resource use efficiency. Land has become a scarce resource, and access to more land implies deforestation and practices that contribute to social and environmental costs [25]. Hence, intensifying food production should be based on increasing productivity per unit of land and per animal [25]. We assume that sustainable intensification of food production relies on enhancing existing input use efficiency to increase the output, which presents a favorable context for the adoption of integrated crop–livestock systems. We measure and compare technical efficiency of different integrated and specialized systems to evaluate this assumption in Nebraska, one of the major beef cattle and crop suppliers in the U.S. and in the world. The state is ranked number one in the country in terms of cattle on feed and beef slaughtering capacity [26]. The value of Nebraska's 2021

field and miscellaneous crops was estimated at USD 16.0 billion, while the value of cattle production was estimated at USD 6.1 billion [27].

Studies comparing the efficiency of specialized and integrated agricultural systems in the U.S. while extending the comparison to the metafrontier do not exist. Our study represents an important step toward understanding the performance of both specialized systems and integrated crop–livestock systems under one production frontier.

There are two major approaches to the measurement and assessment of technical efficiency of farming systems: a parametric analysis such as a stochastic frontier analysis (SFA) or a non-parametric analysis known as data envelopment analysis (DEA). Aigner et al. (1977) and Meeusen and van den Broeck (1977) initiated the stochastic frontier approach based on a parametric econometric model that estimates the stochastic frontier with residuals being decomposed into random error and one-sided error displaying inefficiencies [28,29]. On the other hand, DEA is a non-parametric approach that estimates the production frontier based on linear programming by finding the optimal input-output combination; it was initiated by Färe, Grosskopf, and Kokkelenberg in 1989 [30]. The following presents previous studies applying non-parametric methods to measure technical efficiency of farming systems.

Galluzzo (2018) implemented a DEA model to evaluate the economic performance of farms in Bulgaria [31]. He found that specialized farms with only livestock output (dairy and/or meat) are more efficient than specialized cropping farms. This was explained by the effect of joining the European Union and being under the Common Agricultural Policy (CAP); it was found that the effect of CAP on technical efficiency of farms was very high. A DEA approach was used for the analysis of the technical efficiency of dairy farms in Slovakia between 2006 and 2010 by Michaličkovaá et al. (2013) [32]. Results showed that 96% of farms are technically efficient in producing milk. Technical efficiency was particularly affected by feed costs. Conclusions showed a 4% reduction of feed costs will not have a negative effect on yield and it will enhance the technical efficiency, which implies that the inefficient farms were paying too much for feed. Demircan et al. (2010) assess the technical efficiency of 132 dairy farms in Turkey based on a DEA approach. Results indicate that the sample had an average technical efficiency of 64.2%, while the lowest was 28.6% [33]. The study revealed that forage feed and labor were not used efficiently, and the herd size positively affected the efficiency while land size negatively affected it. The effect of extension services on production efficiency was not significant. Gelan and Muriithi (2010) evaluated the technical efficiency among 371 dairy farms in seventeen districts in east African countries using a DEA model [34]. Eighteen percent of the farms had an efficiency score of 1 which means that they were fully efficient, and they were on the production possibility frontier. Meanwhile, 32% of the farms had an efficiency score below 25% which indicates that they needed to increase their production by 75% while keeping their current level of input fixed to become fully efficient. Findings showed that technology positively affects efficiency of these systems. Veysset et al. (2014) assessed the potential of integrated crop–livestock systems in producing beef sustainably; the authors classified 66 Charolais cattle farms into four groups according to (i) grassland conventional livestock systems, (ii) crop–livestock farms that only sell animal products, (iii) crop–livestock farms that market both crops and beef products, and (iv) organic farms [35]. Integrated farms that sell both beef and crops commodities were found to be less efficient in input use because they miss out on economies of scale. These farms are larger than other farms in other groups and their input use is heavier. Thus, they are not able to translate the environmental and economic benefits of integration into efficiency. This highlights a large gap for research to fill in terms of increasing resource use efficiency and operationalizing the synergies between crop and livestock enterprises.

Our study extends the DEA approach in measuring the technical efficiency of production systems to a metafrontier analysis for the purpose of comparing economic performance of specialized and integrated production systems. In addition, it highlights the role of integration of crops and livestock in promoting sustainable intensification of food production

while shedding light on the areas that need more research and investigation in order to improve integration and achieve balanced tradeoffs.

## 3. Materials and Methods

This study aims to estimate a production frontier using a non-parametric production approach. The non-parametric data envelopment analysis (DEA) uses data on production inputs and outputs to illustrate technical efficiency for homogeneous groups (homogenous in technology) which allows measurement of the performance of farms within that group (Figure 1). Data envelopment analysis employs mathematical linear programming to generate technical efficiency scores, it does not undertake a particular production function, nor does it make assumptions on distributions of a subsequent error term [36].

We use DEA to evaluate technical efficiency levels of different counties in the state of Nebraska defined as either being in group 1 of counties which represents integrated crop–livestock production systems, in group 2 of counties which represents specialized livestock production systems, or in group 3 of counties which represents specialized crop production systems. "The DEA model converts the multiple inputs into multiple outputs to evaluate economic performance through estimating operational processes" [37,38]; it is either input-oriented (minimize inputs while maintaining a constant level of output) or output-oriented (maximize outputs while maintaining a constant level of inputs). In agricultural production efficiency analysis, output-oriented DEA is preferred because the purpose for farmers is predominantly to maximize their profits. Therefore, this study builds an output-oriented DEA where each county is considered to be a decision-making unit (DMU) that produces the maximum feasible dollar value of output for a fixed level of inputs. Every DMU uses different inputs to produce different levels of output and has an efficiency score that is compared to other DMUs efficiency scores [38]. In our case, each county in Nebraska is a collection of farms making up one DMU.

Charnes, Cooper, and Rhodes (1978) introduced a DEA model that presumed constant returns to scale (CRS), identified as the 'CCR model' [39]. Banker, Charnes, and Cooper (1984) extended the CCR model into the 'BCC model' to account for variable returns to scale (VRS) [40]. Our study applies a VRS-DEA model as it encompasses both increasing and decreasing returns to scale. We consider the counties operating as DMUs in this study to be profit maximizers. This gives more flexibility to the data. Suppose we have $N$ decision-making units, each DMU uses K inputs to produce M dollar value of output. The DMU $i$ uses $X_{ki}$ units of the $k$ input to produce $Y_{mi}$ units of the $m$ output. Linear programming methods are used to create a non-parametric piece-wise surface (or frontier) over the data and efficiency measures are calculated relative to this surface [41].

Following Färe, Grosskopf, and Kokkelenberg (1989), the output-oriented DEA LP model maximizing output for a constant level of input is expressed as follows [28]:

$$Max\varnothing_i \qquad (1)$$

$$\varnothing i, Zi$$

$$s.t$$

$$\varnothing_1 Y_{m,i} \leq \sum_i z_i Y_{m,i} \ \ \forall \ m$$

$$\sum_i z_i x_{ki} \leq x_{ki}$$

$$\sum_i z_i = 1$$

where $\varnothing_i$ is the proportional increase in output that county $i$ could attain while maintaining the same level of input. $Y_{m,i}$ is the amount of output $m$ by county $i$, $x_{ki}$ is the amount of input $k$ used by county $i$ and $z_i$ are the weighting factors. The constraint $\sum_i z_i = 1$ allows for variable returns to scale. The output-oriented technical efficiency score of county $i$ is defined as the ratio of observed output to the efficient output in Equation (2) [42]:

$$TE = \frac{Y_i}{Y_i^*} = \frac{Y_i}{\phi_i Y_i} = \frac{1}{\phi_i} \tag{2}$$

One of the limitations of non-parametric methods such as data envelopment analysis in measuring productivity and technical efficiency is that they only require deterministic indicators and do not take into consideration random statistical noise [43]. Only a set of inputs and outputs that are controlled by the farmer are considered.

The metafrontier approach measures the technical efficiency of each DMU taking into consideration the heterogeneity of production technology, scale, type, and other inherent characteristics among the studied sample. It was first proposed by Hayami (1969) and Rutten and Hayami (1973) [44,45]. For the current study, we have heterogeneous production systems resulting in three groups of counties classified by type of production system. Each group of counties forms a group frontier and, consequently, the new production frontier is formed through enveloping all three frontiers and forming the metafrontier (Figure 1). The metafrontier concept was applied in multiple studies and by several researchers to measure efficiency in different areas [46].

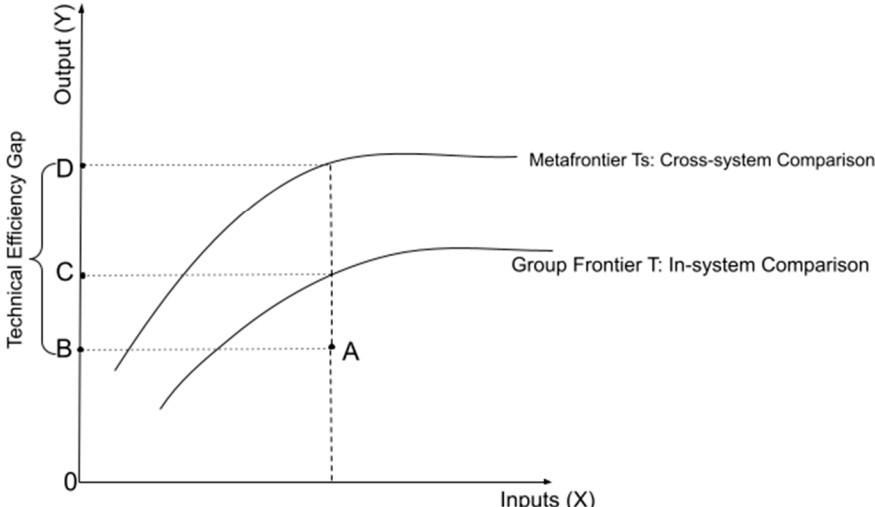

**Figure 1.** Graphic illustration of metafrontier approach: ((X) is the axis presenting aggregate input data and (Y) is the axis for one output data. A is the actual input-output combination of the county and curve T is the group frontier joining optimal input-output arrangements. [BC] is the inefficiency of A compared to the group frontier for in-system comparison and [BD] is the technical efficiency gap which presents the inefficiency of A compared to the metafrontier Ts. [CD] is the inefficiency of the group compared to the metafrontier for cross-system comparison). Own elaboration based on Bianchi et al., (2020) [47].

- Typology of the different farming systems in Nebraska:

The state of Nebraska consists of 93 counties and exists in the Midwestern region of the United States (Figure 2). It is bordered to the east by Iowa and to the southeast by Missouri. Wyoming borders Nebraska to the west, Kansas to the south and Colorado to the southwest. Finally, South Dakota borders Nebraska in the north. The state measures over 200,000 km$^2$ and it consists of two main land regions. The Dissected Till Plains cover the gently rolling hills in the east. Meanwhile, the Great Plains occupy most of the western part of the state and consist of smaller land regions including the Sandhills, the Pine Ridge, the Rainwater Basin, the High Plains, and the Wildcat Hills. Average annual precipitation decreases from 800 mm in the east to 350 mm in the west. Agricultural activities, cropping, and cattle production are the primary economic drivers of the state's economy.

The three types of production systems investigated in this study are specialized livestock systems, specialized cropping systems and integrated crop–livestock systems; the

grouping of counties based on the type of dominant production system was completed based on an economic determinant characterizing the contribution of livestock to the farm income (share of livestock sales in total sales in % [SLS]). Ultimately, we defined a production system according to the monetary output of the different activities present in each county.

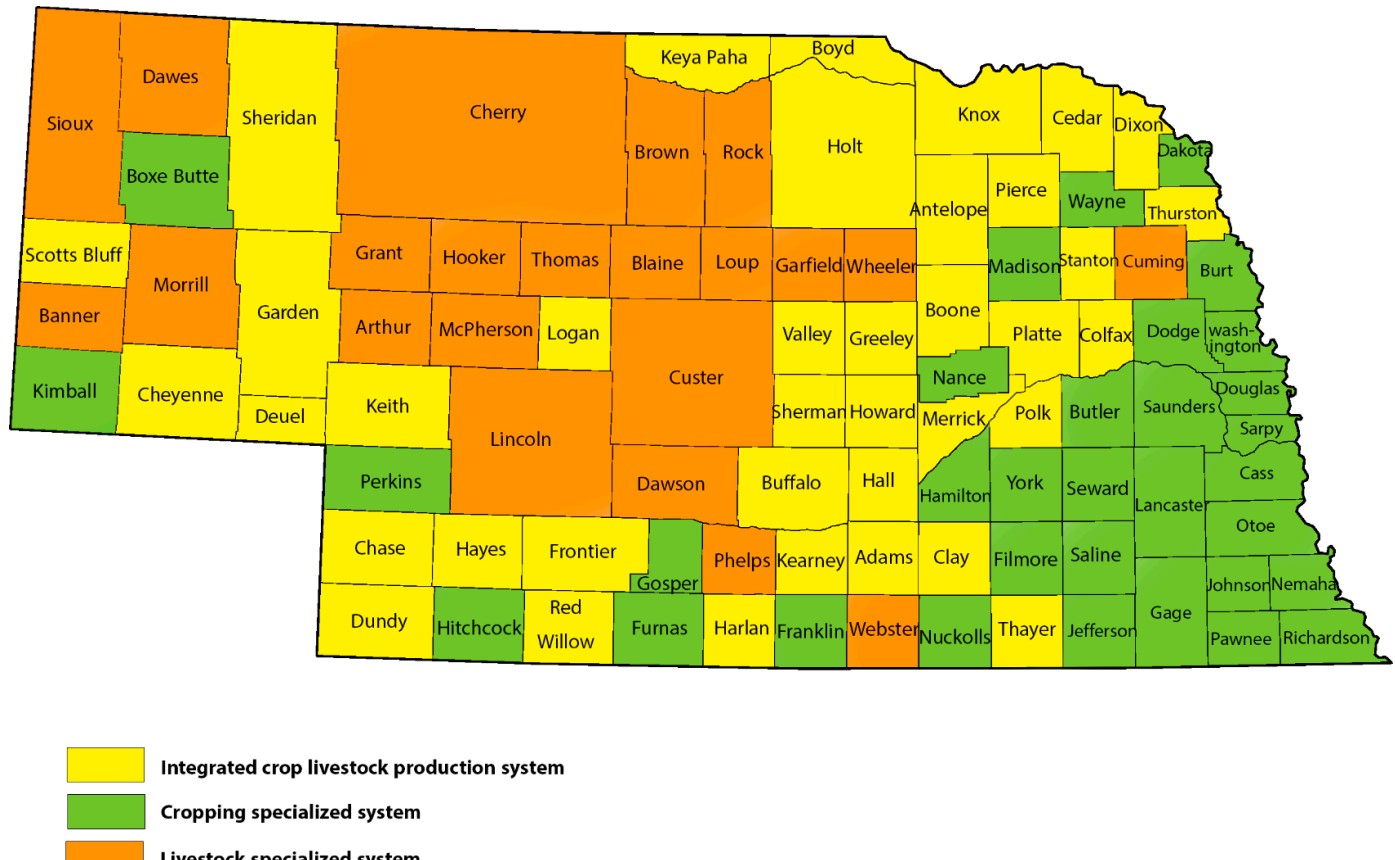

**Figure 2.** Presentation of dominant agricultural production systems in Nebraska counties.

The typology methodology used in this study involved both statistical analysis and expert judgment. The spatial scale of this analysis was not farm-level. The analysis was conducted at the regional level using aggregate county-level data. We used the variable SLS to classify the different agricultural systems in each county. Since every county should have some livestock, our economic determinant requires some judgement of the type of production system to assign. In this study, we want to specify that when we refer to integrated crop–livestock systems, we refer only to cattle livestock, and we do not include other animal products in the analysis. In summary, the study used both statistical analysis and expert judgment to develop a typology of agricultural systems based on the share of livestock sales in total sales in each county (Table 1). It is worth noting that no observation was found where the SLS was between 60% and 70%. This natural split in the data led to the choice of SLS $\geq$ 70% as the threshold to classify specialized livestock systems. Likewise, another natural split developed at SLS $\leq$ 40% for classifying specialized cropping systems. Our aim was to capture heterogeneity across different Nebraska regions and specific production features of each county. We considered specific features of each county including natural capital (type of land, major crops, share of pastureland, and cattle inventory) to determine the type of production system represented. This approach, along with the chosen thresholds, contributed to the classification of ICLS as counties with 40% < SLS < 70% to ensure a robust typology.

**Table 1.** Classification of counties and farming systems.

| Share of Livestock Sales in Total Sales in % (SLS) | Type of Production System | Number of Counties |
|---|---|---|
| SLS ≥ 70% | Specialized livestock | 22 |
| 40% < SLS < 70% | Integrated crop–livestock | 38 |
| SLS ≤ 40% | Specialized cropping | 33 |

- Data:

This study applies cross-sectional aggregate county data from the United States Department of Agriculture (USDA) census collected in 2017 [26]. The census is an important source of information on the production management practices, resource use, and economic well-being of America's farms and ranches. Table 2 presents descriptive statistics for the inputs and output. Our first analysis using DEA to produce group frontiers incorporates inputs and outputs that are not identical for the three systems, the two specialized systems, and the integrated crop–livestock system. All inputs and outputs are computed in terms of monetary unit (USD) except for land area and number of cattle heads. First, for the specialized cropping system, inputs include agricultural land area in hectares, chemical inputs (fertilizers + pesticides) in USD, and labor in USD. For the specialized livestock system inputs are number of cattle heads, feed costs, labor, and agricultural land area. Finally, for the integrated crop–livestock system, the inputs are total agricultural land, herd size, total labor (livestock + crops), and feed costs (off-farm and in-farm) in addition to chemical inputs (fertilizers and pesticides). The output in all systems is total net income from both crops and livestock enterprises; the sample size consists of 93 counties of Nebraska.

**Table 2.** Descriptive statistics for the inputs and the output of production systems.

| | | | | Inputs | | | Output |
|---|---|---|---|---|---|---|---|
| | | Chemicals [1] | Labor [1] | Land [2] | Cattle [3] | Feed [1] | Net Income |
| Integrated crop–livestock systems | Mean | 25,307,605 | 8,667,789 | 123,543 | 8825 | 39,464,131 | 47,038,921 |
| | Standard Error | 1,948,025 | 794,067 | 24,425 | 6820 | 5,263,547 | 4,621,908 |
| | Minimum | 3,988,000 | 1,553,000 | 31,235 | 17,345 | 2,053,000 | 4,899,000 |
| | Maximum | 56,540,000 | 24,478,000 | 315,680 | 175,123 | 171,620,000 | 125,163,000 |
| Specialized cropping systems | Mean | 25,767,363 | 7,925,757 | 82,168 | | | 44,072,606 |
| | Standard Error | 1,712,118 | 675,913 | 20,605 | | | 4,195,104 |
| | Minimum | 7,528,000 | 1,258,000 | 14,370 | | | 7,406,000 |
| | Maximum | 42,565,000 | 16,119,000 | 212,872 | | | 83,831,000 |
| Specialized livestock systems | Mean | | 8,830,380 | 154,274 | 150,838 | 56,304,666 | 39,379,333 |
| | Standard Error | | 1,766,159 | 71,026 | 33,953 | 14,493,456 | 9,453,731 |
| | Minimum | | 1,237,000 | 35,236 | 19,913 | 3,583,000 | 2,796,000 |
| | Maximum | | 24,827,000 | 479,017 | 568,098 | 223,038,000 | 167,517,000 |

[1] The inputs are chemicals, labor, and feed, and the output net incomes are in USD. [2] The input land is in hectares. [3] The input for cattle is in number of heads.

## 4. Results and Discussion

Nebraska farms and ranches utilize 18.3 million hectares, equivalent to 92% of the state's total land area [48]. The cropped land constitutes 8 million hectares, of which 75% is utilized to grow corn and soybeans [49]. There are more than 9 million hectares of ranches and pastureland [50]; half of which are in the Sandhills grassland region in the north central part of the state. The total number of beef cattle in the state is 1.8 million head, while Nebraska also feeds around 5 million head of cattle each year in Nebraska feed yards, with cattle coming into the state from all over the United States. Ultimately, the major economic activity in the state is agriculture and Nebraska is a major contributor to food supply in the U.S. and in the world. Hence, the generosity of resources and diversity of landscape is emphasized in a diversification of farming systems across the state. Following

our typology, we classified each of the 93 counties in Nebraska into one of three types according to the omnipresent agricultural production system in the county (Table 1 and Figure 2). We classified 22 counties, or 24%, as specialized livestock systems where 70% or more of the total sales are livestock sales from cattle herds. Specialized cropping systems counties represent 35% of the total number of counties, equivalent to 33 counties, where crop sales provide more than 60% of the total sales. Finally, 38 counties with livestock cattle sales supplying more than 40% and less than 70% of the total sales were classified as the integrated crop–livestock systems counties (41%).

### 4.1. In-System Comparison for Specialized Livestock Counties

We calculated the average level of technical efficiency among livestock specialized counties as 75%, showing the potential to increase efficiency by as much as 25% on average. We classified eight counties that specialized in livestock production (36%) as fully efficient with a technical efficiency score equal to 1 under the group efficiency analysis (Table 3).

**Table 3.** In-system comparison: Technical efficiency scores distribution for counties under each type of farming system.

|  | Integrated Crop–Livestock Systems | | Specialized Livestock Systems | | Specialized Cropping Systems | |
|---|---|---|---|---|---|---|
| **Mean Efficiency** | 80% | | 75% | | 80% | |
| **Technical Efficiency** | **Number of Counties** | **%** | **Number of Counties** | **%** | **Number of Counties** | **%** |
| TE = 1 | 7 | 18 | 8 | 36 | 10 | 30 |
| TE < 1 | 31 | 82 | 14 | 64 | 23 | 70 |
| Total | 38 | 100 | 22 | 100 | 33 | 100 |

Cuming County is one of the fully efficient livestock specialized counties, with a relatively small agricultural land area and a high number of cattle per hectare ratio equal to 8.01. Cuming County is located in eastern Nebraska where crop production, particularly corn and soybean, is dominant. Many cattle in Cuming County are being fed in feed yards, which explains the reduced use of fertilizers and other chemicals per dollar value of output. Cuming County is ranked first in agricultural sales in Nebraska, of which 87% are from livestock production activities, mainly cattle feeding [51]. Feedlots may create pressure on the environment and threaten the sustainability of food production in the county, the state, and the nation [52]. Livestock waste can be a valuable source of crop nutrients when appropriately managed and feedlots are a rich source of manure; thus, it creates opportunities for integrating crops and livestock, which on one hand reduces ICLS production costs and, on the other hand, provides alternative feeding sources for cattle [53–55]. Cherry, Grant, and Hooker Counties are other examples of fully efficient livestock specialized counties. They are all Sandhills grassland counties with large ranches that graze cattle, and they have very low expenditures on chemicals. The Sandhills are grazing livestock systems with sandy soil structure, low precipitation, and high evaporation rates which undermines the possibility of integrating crops into the existing livestock systems. McPherson and Banner Counties are the least efficient counties of the specialized livestock system group with efficiency scores of 0.45 and 0.42, respectively. McPherson County measures an agricultural area of 105,000 hectares which is larger than Cuming and Banner counties (70,000 hectares and 83,000 hectares, respectively); however, the number of cattle is only about 23 k, which is about half the number of cattle in Banner and only 4% of what is in Cuming County. The sandy structure of McPherson soils does not give a large opportunity to extend crop farming in the area, but an increase of the number of cattle could help enhance the efficiency of the county. Cumming et al. (2019) examined Nebraska rangeland capacity for cattle production; the grazing efficiency score for McPherson County was 20% below the state average indicating the potential for increasing efficiency through improved grazing management practices [56].

Banner County is on the western side of the state; topographically, it discloses better opportunities for integrating crops and livestock. However, adopters should optimize the labor allocation between the two activities. Banner County is not very populated, and the agricultural labor force is reduced, leading to opportunities as well as challenges for integrating more crops into production systems aiming to increase efficiencies.

### 4.2. In-System Comparison for Specialized Cropping Counties

The mean technical efficiency score registered in the group of cropping specialized counties was 80% (Table 3). Overall, we classified 10 counties (30%) out of 33 of the cropping specialized counties as fully efficient (see Table A1). They spend an average of around USD 26 million on chemical inputs. Hitchcock County, in the southwest part of the state, has the lowest efficiency score at 0.42; a small number, especially for a county whose economy is strongly dependent on agriculture. Two major issues seem to be affecting the efficiency score for this county. The first is that only 11% of its cropped land is under irrigation. The second is the low expenditures on chemicals (USD) per hectare of cropped land. Hitchcock County averaged only USD 160.55 per hectare on chemical inputs. Meanwhile, the average expenditure on chemical inputs for the efficient cropping specialized counties is USD 330.40 per hectare. Socioeconomic conditions (rural poverty, high risks, etc.) may contribute to the low chemical input expenditures in Hitchcock County.

Most of the cropping specialized counties are agglomerated in the eastern and southeastern part of the state where precipitation is higher. Corn and soybean rotations are widely practiced among growers in this area of the state [52] which undermine crop diversity and engender pest outbreaks and nitrogen runoff [54]. Our results indicate that the most technically efficient production systems have higher chemical costs and thus tend to be less environmentally efficient. A study by Kladivco et al. (2014) stated that adoption of cover crops in the U.S. Midwest could decrease 20% of $NO_3$ in the Mississippi River [57]. Eastern and southeastern Nebraska growers have the potential to be more environmentally efficient if they incorporate cover crops in their rotation, and thus reduce soil erosion and weed invasion which will result in less expenditures on chemicals. Cover crops will also provide feed sources to cattle in the highlighted counties and diversify income sources. Therefore, adopting cover crops constitutes an alternative to promote the integration of cattle production in Eastern Nebraska and an opportunity for sustainable intensification of food production. The integration of crops and livestock in Eastern Nebraska will also maximize energy yields; raw materials of cattle and biomass from corn give the state the potential to be a leader in expanding biofuel usage and producing clean energy [58].

### 4.3. In-System Comparison for Integrated Crop–Livestock Counties

Nebraska has 38 counties that we classified as ICLS counties, with cattle sales in these counties constituting between 40% and 70% of all agricultural sales. The mean technical efficiency score among the ICLS group of counties is 80%, but only 18% of the counties are fully efficient (Table 3). Results showed that the main factors affecting efficiency in these ICLS counties are primarily the chemical expenditures per hectare and the feed costs per head of cattle. Findings also show that the share of irrigated cropping land and the share of pastureland in fully efficient counties is higher than those with lower technical efficiency scores. Fully efficient ICLS counties spend less on chemicals, but they have higher feed costs per head. On average, they spend USD 172.90 per hectare on pesticide and fertilizers and around USD 470 per head on animal feed. Crop–livestock integration relies on synergies between the two systems where the crop residues are used as animal feed; Polk County is a fully efficient ICLS county, and it presents an example of good management of resources between the animal and cropping enterprises. The county has a large sized agricultural area, of which 72% is irrigated land, the total agricultural sales are distributed equally between livestock and cropping products, corn and soybean are the dominating crops, and crop residue grazing is an important source for animal feed. Furthermore, Polk County is among the counties with the highest estimated grazing efficiency score (46%) [55]. This

indicates a high number of beef cattle grazing in the county relative to supply (AUMs) from perennial grass and may be attributed to more intense perennial grass grazing systems and more extensive use of cropland residue as a grazing resource.

### 4.4. Cross-System Comparison: Metafrontier Technical Efficiency

The metafrontier is the production frontier that envelops all three groups' frontiers and allows the comparison between the heterogenous production systems. The average technical efficiency registered among all 93 counties is 74% (Table 4). We note that 23 counties make up the metafrontier (see Table A1). Of these counties, 57% (13) represent specialized cropping systems, 30% (7) represent specialized livestock systems, and only 13% (3) represent ICLS counties. Overall, 35% (33) of the 93 counties in Nebraska were classified as specialized cropping systems, 24% (22) were classified as specialized livestock systems, and 41% (38) were classified as ICLS counties. ICLS counties tended to have higher total expenses than specialized systems counties, coming from higher chemical costs, higher feed costs, and higher labor costs. Integrated crop–livestock systems may perform best if complementarities between the two subsystems can be taken advantage of, such as crop residue grazing. For example, Polk County was on the efficiency frontier for the ICLS counties as well as the metafrontier thanks to efficient use of its grazing capacity. Logan County in the Sandhills region and Platte County in east central Nebraska were the other two ICLS counties on the metafrontier. Logan County relies on efficient grazing systems with some irrigated cropland while Platte County relies on irrigated cropland complementing feedlots which reduce their feed expenses remarkably.

**Table 4.** Cross-systems comparison: Metafrontier technical efficiency scores distribution for three types of farming systems.

| | Mean TE | Standard Deviation | Number of Counties with TE = 1 per Type of Production System |
|---|---|---|---|
| Integrated crop–livestock systems | | | 3 (7%) |
| Cropping specialized systems | 74% | 0.24 | 13 (39%) |
| Livestock specialized systems | | | 7 (32%) |

Table 4 shows that 39% of specialized cropping system counties and 32% of specialized livestock system counties have a technical efficiency score equal to 1 on the metafrontier. Specialized cropping systems have a better technical efficiency capacity when compared to the specialized livestock or to the ICLS counties due to the high expenditures on feed for the cattle in both livestock production systems. There is an important opportunity for integrated systems to achieve higher technical efficiency from optimally using possible synergies and complementarities that could lower feed costs under the umbrella of integration. The utilization of crop residues for grazing, and cover crops could reduce feed costs and improve efficiencies regarding the animal presence in the system [22]. The ICLS diversifies income sources and provides manure. Integration of crops and livestock relies on three possible associations: (1) the incorporation of perennial vegetation, (2) the adoption of cover crops providing higher quality animal feed, and (3) grazing crop residues [58]. The first two methods of coupling crops and livestock are associated with environmental and economic benefits. Residue grazing is also economically beneficial and remains unharmful to the soils if it is rationally practiced [59]. The least technically efficient integrated counties are those with higher labor costs and higher chemical costs. We should consider that in integrated systems, one unit of land input is used as an input for both production activities which increases land use efficiency [53]. Some synergies and balances should be put in place in the use of other inputs such as feed, chemicals, etc. [6,59]. Future research should create general settings for a beneficial and successful integration taking into consideration the value of diversification in controlling risk, which was not included in the present study, and which is an important component in sustainability studies.

Although this study is a crucial first step towards understanding the determinants of technical efficiency in integrated crop–livestock systems, compared to specialized systems, it is necessary to note that our work was constrained by limited access to data. While our research provides valuable insights into the effect of resource management and input usage on technical efficiency at a regional level, the scope of this study was limited to aggregate county-level data. Thus, we could not conduct a more detailed farm-level analysis that could capture more accuracy for identifying farming systems and selecting determinants of technical efficiency that included exogenous factors. Despite these limitations, our findings emphasize regional differences in production systems and capacities and lay the foundation for future research examining the effect of exogenous variables on technical efficiency. Future research could consider exogeneity in terms of demographic, political, socio-economic, and environmental determinants of technical efficiency in different farming systems and different agricultural contexts. A more comprehensive examination of farm-level data would deepen our understanding of the potential of integrated crop–livestock systems in promoting sustainable intensification of food production in Nebraska and beyond.

## 5. Conclusions and Implications

Integrated crop–livestock systems are promoted as a strategic alternative for sustainable intensification, based on multiple natural synergies and complementarities that could take place between the two systems. This paper focuses on classifying counties in the state of Nebraska into three groups based on the dominating farm production system. It uses a non-parametric data envelopment approach to measure the technical efficiency of the different groups. The DEA model utilized is an output-oriented model under variable returns to scale where we assume that farmers are profit maximizers. It was applied to 2017 aggregate county-level data for Nebraska that was obtained from the USDA Census of Agriculture. Nebraska is known as one of the largest agricultural states in the U.S., it has a large difference in soil types and precipitation across the different regions, which creates remarkable diversity in agricultural activity. We distinguished three different production systems depending on the contribution of livestock to total agricultural sales, 38 counties were classified as integrated crop–livestock systems while the rest were classified as specialized counties, 33 are specialized in cropping and 22 specialized in livestock production. Results show that the number of cattle and feed costs are the main determinants of technical efficiency in specialized livestock systems where the mean technical efficiency is 75%. Fully efficient livestock specialized counties are those who have feedlots for efficient animal feeding or productive grasslands for cost efficient grazing. Meanwhile, specialized livestock systems counties with low efficiency scores have a very high feed cost per head ratio. Specialized cropping systems are present in 33 counties across Nebraska, and they have an average technical efficiency of 80%. The major determinant of technical efficiency in these systems is the expenditure on chemicals with an average ratio among efficient counties of USD 330.40 per hectare.

The average technical efficiency of the integrated crop–livestock systems counties in the in-system comparison was also 80%. However, the integrated crop–livestock counties have lower efficiency compared to the specialized cropping and specialized livestock systems under the metafrontier. The metafrontier is a production boundary that encompasses the frontiers of all the three groups, enabling the comparison of diverse production systems. The average technical efficiency among all 93 counties for the metafrontier analysis is 74%. A total of 23 counties from different production systems operate at full efficiency and are placed on the metafrontier; of which 57% are specialized cropping systems counties, 30% specialize in livestock production, and only 13% (equivalent to 3 counties) are integrated crop–livestock systems. We found that labor costs are higher in the inefficient ICLS counties in addition to having relatively higher expenses in terms of feed and chemicals. We discuss a potentially valuable integration between crops and livestock in eastern Nebraska where the dominant agricultural production system is a cropping system consisting mainly of corn

and soybeans. However, there are many constraints to integration in the pasture and range dominated Sandhills of north central Nebraska, including soil structure and vegetation type. For crop–livestock integration to be beneficial, managerial skills are needed in order to understand the tradeoffs and maintain balanced synergies between the two enterprises. A successful integration, at farm-level and regional-level equally, requires the coexistence of multiple social, political, natural, and economic factors.

This study is a first step in understanding the determinants of technical efficiency of integrated crop–livestock systems compared to specialized agricultural systems. It helps preview the direct effect of resource management on technical efficiency. It also highlights the differences between production regions in terms of production systems and capacities in meeting full technical efficiency. It provides a foundation for discussion of the potential for sustainable intensification of food production through enhancing resource use efficiency. Future extensions of this research include examining the effect of exogenous variables on technical efficiency and introducing demographic, political, socio-economic, and environmental factors for a better understanding of the determinants of technical efficiency in different farming systems. A more detailed look into farm-level data will increase understanding of the role integrated crop–livestock systems play in contributing to sustainable food intensification and food security in Nebraska, in the United States, and in the world.

**Author Contributions:** Conceptualization, M.A. and J.P.; Methodology, M.A.; Formal analysis, M.A.; Data curation, M.A.; Writing—original draft, M.A.; Writing—review & editing, J.P.; Supervision, J.P.; Project administration, J.P.; Funding acquisition, J.P. All authors have read and agreed to the published version of the manuscript.

**Funding:** This research was supported by the U.S. Department of Agriculture, Economic Research Service and cooperative agreement 58-6000-1-0073. The findings and conclusions in this publication are those of the author(s) and should not be construed to represent any official USDA or U.S. Government determination or policy. The funder website is https://www.ers.usda.gov/. The funders had no role in study design, data analysis, decision to publish, or preparation of the manuscript.

**Institutional Review Board Statement:** Not applicable.

**Informed Consent Statement:** Not applicable.

**Data Availability Statement:** The data source for this information can be found in the USDA National Agricultural Statistics Service's Quick Stats, which is available online at https://data.nal.usda.gov/dataset/nass-quick-stats. The data was accessed on 15 March 2022.

**Acknowledgments:** We thank Richard Perrin and Lilyan Fulginiti, professors in the Department of Agricultural Economics, University of Nebraska-Lincoln, for providing insight and expertise that greatly assisted this research.

**Conflicts of Interest:** The authors declare no conflict of interest. The funders had no role in the design of the study; analyses, or interpretation of data; in the writing of the manuscript; or in the decision to publish the results.

## Appendix A

**Table A1.** Presentation of counties, farming system type, group technical efficiency score, and metafrontier technical efficiency score.

| County | Farming System Type | TE (Groups) | TE-Metafrontier |
|---|---|---|---|
| Custer | Specialized livestock | 0.56 | 0.49 |
| Dawson | Specialized livestock | 0.81 | 0.66 |
| Blaine | Specialized livestock | 0.64 | 0.83 |
| Cherry | Specialized livestock | 1.00 | 1.00 |
| Garfield | Specialized livestock | 0.19 | 0.18 |
| Loup | Specialized livestock | 1.00 | 1.00 |

**Table A1.** *Cont.*

| County | Farming System Type | TE (Groups) | TE-Metafrontier |
|--------|---------------------|-------------|-----------------|
| McPherson | Specialized livestock | 0.45 | 0.56 |
| Rock | Specialized livestock | 0.70 | 0.57 |
| Thomas | Specialized livestock | 0.73 | 0.91 |
| Wheeler | Specialized livestock | 0.67 | 0.96 |
| Cuming | Specialized livestock | 1.00 | 1.00 |
| Banner | Specialized livestock | 0.42 | 0.28 |
| Dawes | Specialized livestock | 0.61 | 0.48 |
| Morrill | Specialized livestock | 1.00 | 0.72 |
| Sioux | Specialized livestock | 1.00 | 1.00 |
| Lincoln | Specialized livestock | 0.88 | 0.76 |
| Arthur | Specialized livestock | 0.46 | 0.58 |
| Brown | Specialized livestock | 0.78 | 1.00 |
| Phelps | Specialized livestock | 1.00 | 0.74 |
| Webster | Specialized livestock | 0.62 | 0.44 |
| Grant | Specialized livestock | 1.00 | 1.00 |
| Hooker | Specialized livestock | 1.00 | 1.00 |
| Buffalo | Crop–livestock Integrated | 0.97 | 0.81 |
| Hall | Crop–livestock Integrated | 1.00 | 0.83 |
| Howard | Crop–livestock Integrated | 0.54 | 0.45 |
| Sherman | Crop–livestock Integrated | 0.72 | 0.49 |
| Valley | Crop–livestock Integrated | 0.64 | 0.51 |
| Colfax | Crop–livestock Integrated | 1.00 | 0.98 |
| Platte | Crop–livestock Integrated | 1.00 | 1.00 |
| Polk | Crop–livestock Integrated | 1.00 | 1.00 |
| Boyd | Crop–livestock Integrated | 1.00 | 0.99 |
| Holt | Crop–livestock Integrated | 0.83 | 0.77 |
| Keya Paha | Crop–livestock Integrated | 1.00 | 0.65 |
| Logan | Crop–livestock Integrated | 1.00 | 1.00 |
| Antelope | Crop–livestock Integrated | 0.88 | 0.84 |
| Boone | Crop–livestock Integrated | 0.97 | 0.87 |
| Cedar | Crop–livestock Integrated | 0.71 | 0.67 |
| Dixon | Crop–livestock Integrated | 1.00 | 0.93 |
| Knox | Crop–livestock Integrated | 0.82 | 0.68 |
| Pierce | Crop–livestock Integrated | 1.00 | 0.77 |
| Stanton | Crop–livestock Integrated | 0.64 | 0.51 |
| Thurston | Crop–livestock Integrated | 1.00 | 0.96 |
| Cheyenne | Crop–livestock Integrated | 0.54 | 0.44 |
| Deuel | Crop–livestock Integrated | 1.00 | 0.26 |
| Garden | Crop–livestock Integrated | 0.65 | 0.52 |
| Scotts Bluff | Crop–livestock Integrated | 0.76 | 0.53 |
| Sheridan | Crop–livestock Integrated | 1.00 | 0.83 |
| Adams | Crop–livestock Integrated | 0.75 | 0.71 |
| Harlan | Crop–livestock Integrated | 1.00 | 0.77 |
| Kearney | Crop–livestock Integrated | 0.65 | 0.62 |
| Clay | Crop–livestock Integrated | 1.00 | 0.78 |
| Thayer | Crop–livestock Integrated | 0.84 | 0.56 |
| Dundy | Crop–livestock Integrated | 0.80 | 0.58 |
| Frontier | Crop–livestock Integrated | 0.13 | 0.10 |
| Hayes | Crop–livestock Integrated | 0.41 | 0.34 |
| Keith | Crop–livestock Integrated | 0.51 | 0.39 |
| Red Willow | Crop–livestock Integrated | 0.37 | 0.32 |
| Greeley | Crop–livestock Integrated | 0.75 | 0.60 |
| Merrick | Crop–livestock Integrated | 0.76 | 0.60 |
| Chase | Crop–livestock Integrated | 0.72 | 0.66 |
| Butler | Specialized crops | 0.78 | 0.97 |
| Cass | Specialized crops | 1.00 | 1.00 |
| Dodge | Specialized crops | 1.00 | 1.00 |

**Table A1.** *Cont.*

| County | Farming System Type | TE (Groups) | TE-Metafrontier |
| --- | --- | --- | --- |
| Douglas | Specialized crops | 1.00 | 1.00 |
| Hamilton | Specialized crops | 0.97 | 1.00 |
| Lancaster | Specialized crops | 0.86 | 0.96 |
| Nance | Specialized crops | 0.68 | 0.47 |
| Sarpy | Specialized crops | 1.00 | 1.00 |
| Saunders | Specialized crops | 0.86 | 0.81 |
| Seward | Specialized crops | 1.00 | 1.00 |
| Washington | Specialized crops | 1.00 | 1.00 |
| York | Specialized crops | 0.93 | 0.88 |
| Burt | Specialized crops | 0.99 | 0.91 |
| Dakota | Specialized crops | 0.81 | 1.00 |
| Madison | Specialized crops | 0.86 | 0.79 |
| Wayne | Specialized crops | 0.75 | 0.72 |
| Box Butte | Specialized crops | 0.52 | 0.46 |
| Kimball | Specialized crops | 1.00 | 0.57 |
| Franklin | Specialized crops | 0.46 | 0.69 |
| Furnas | Specialized crops | 0.38 | 0.40 |
| Gosper | Specialized crops | 0.27 | 0.24 |
| Fillmore | Specialized crops | 0.82 | 0.97 |
| Gage | Specialized crops | 0.81 | 1.00 |
| Jefferson | Specialized crops | 0.60 | 0.65 |
| Johnson | Specialized crops | 1.00 | 0.96 |
| Nemaha | Specialized crops | 0.73 | 1.00 |
| Nuckolls | Specialized crops | 0.70 | 0.94 |
| Otoe | Specialized crops | 0.96 | 1.00 |
| Pawnee | Specialized crops | 1.00 | 1.00 |
| Richardson | Specialized crops | 1.00 | 1.00 |
| Saline | Specialized crops | 0.69 | 0.85 |
| Hitchcock | Specialized crops | 0.42 | 0.56 |
| Perkins | Specialized crops | 0.67 | 0.93 |

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
