# Peer review of "Integrated vs. Specialized Farming Systems for Sustainable Food Production: Comparative Analysis of Systems’ Technical Efficiency in Nebraska"

_sustainability, doi:10.3390/su15065413_

Round 1

Reviewer 1 Report

Dear Authors, 

Proposed study entitled" Integrated vs. Specialized Farming Systems for Sustainable Food Production: Comparative Analysis of Systems Technical Efficiency in Nebraska is very impressive but need some improvements. Can you please follow some suggested points and revised accordingly. 

Abstract: 

1-General introduction is enough but suggested to add brief methodology

2- Compare the values of Integrated vs. Specialized Farming Systems in % age

3-  Analysis of Systems Technical Efficiency should be included in abstract with interventions. 

2- Introduction: 

1- The novelty of work is missing. Add at end of the introduction

2- Why the current area is chosen for the study? Explain in the introduction.

3- How integrated vs Specialized Farming system helps in achieving sustainability goal should be explained at least half of the page in Introduction Chapter. 

3- Methodology: 

1- The figure 1 is of low quality. Supply all figures of high quality. 

2- Methodology analysis techniques are not clear please improve it 

4- Results: 

1- In all the results and measurement mentioned in the table does not have error values? Ensure the error obtained

2- Improve the conclusion with few more information

3-In discussion section the citation of literature is missing in many places

4- The limitation of the work is not explained. Explain at least half of the page before the conclusion. 

5- There is lot of grammatical and sentence error. Check it throughout

5 References: 

Check out all the references according to the MDPI format (APA)

Author Response

Please see the attachment. Thank you for your review comments and suggestions. They greatly improved our article.

Reviewer 2 Report

I give the following three suggestions for this article:

1. First of all, the definition of production system in the article is vague. What is the basis for classification of the three production systems? Need further explanation from the author.

2. In the result section, it is not enough to compare the production efficiency of three different production systems only according to the results of DEA model. Can you further explain and compare the degree of input relaxation of each group?

3. In the part of literature introduction, it is necessary to supplement the literature on the exploration of different mechanisms for the production efficiency of the three production systems to provide evidence for the later research.

Author Response

(The authors gave the same response as above.)

Reviewer 3 Report

This paper assesses the technical efficiency of integrated crop livestock systems compared to specialized cropping and specialized livestock systems in the state of Nebraska, in the Midwestern region of the United States based on the 2017 aggregate census data. The author found that counties representing integrated crop-livestock systems are less likely to appear on the metafrontier. It is suggested the need for further research on optimal crop-livestock integration for food production.

In my opinion, this paper addresses an important but underexplored agricultural question. However, I have some reservations regarding the identification strategy. This needs some minor corrections are stated as follows:

i.            The author wrote that “we classified each of the 93 counties in Nebraska into one of three 319 types according to the omnipresent agricultural production system in the county (Table 1 320 & figure 2). We classified 22 counties or 24% as “specialized livestock systems” as 70% or 321 more of the total sales are livestock sales from cattle herds. “

a)        What are the criteria based on?

b)       The reasonableness of this division should be explained.

ii.            As we know that among the crop-livestock systems, there are differences in input-output between planting industry and breeding industry, should weight be given according to the structure, and then conversion summary of input-output?

iii.            Whether pesticides should be considered for inputs?

Author Response

(The authors gave the same response as above.)
